# Polar Bloch points in strained ferroelectric films

Yu-Jia Wang [1,8], Yan-Peng Feng [2,3,8], Yun-Long Tang [1], Yin-Lian Zhu [2,3,4], Yi Cao[1,5], Min-Jie Zou[2,3], Wan-Rong Geng[2,3] & Xiu-Liang Ma [2,3,6,7] ✉

Topological domain structures have drawn great attention as they have potential applications in future electronic devices. As an important concept linking the quantum and classical magnetism, a magnetic Bloch point, predicted in 1960s but not observed directly so far, is a singular point around which magnetization vectors orient to nearly all directions. Here we show polar Bloch points in tensile-strained ultrathin ferroelectric $PbTiO_3$ films, which are alternatively visualized by phase-field simulations and aberration-corrected scanning transmission electron microscopic imaging. The phase-field simulations indicate local steady-state negative capacitance around the Bloch points. The observation of polar Bloch points and their emergent properties consequently implies novel applications in future integrated circuits and low power electronic devices.

Real-space topologically-protected domain structures in magnets and ferroelectrics have attracted diverse interest in condensed matter physics. Since the prediction of flux-closure domain structures in magnetic films in 1940s (ref. [1,2]), various topological magnetic domain structures have been reported in magnetic films and nanostructures[3–11]. A topological magnetic Bloch point is a singular point around which the magnetization vectors orient to nearly all directions. It was first predicted as a node of a Bloch line on a Bloch domain wall, that is, a Bloch point pinches the Bloch wall and Bloch line to a singular point[12–17]. Recently, magnetic Bloch points attracted more and more interest since they naturally link classical and quantum magnetism[18,19]. They were found to play important roles in many interesting topological transitions, such as the switching of vortex[20–22], the vortex-antivortex annihilation[23], and the unwinding of a skyrmion lattice[24], etc. After half-a-century's efforts, the internal structure of a magnetic Bloch point has not been directly observed experimentally[25].

Due to the similarity between ferromagnets and ferroelectrics, many topological domain structures have been discovered or predicted in ferroelectrics recently, such as flux-closures[26], vortices[27], bubbles[28], skyrmions[29], merons[30], center domains[31,32], and hopfions[33]. The sizes of these ferroelectric domains are generally smaller than their magnetic counterparts, thus holding promise for high-density information storage, thin-film capacitors, actuators and other electronic devices. Recently, Bloch points have been predicted in several polar systems. For example, Salje and Scott studied the polar domain walls in $SrTiO_3$ and found Bloch lines and Bloch points could exist on these polar domain walls[34]. Morozovska et al. studied the domain structures of a $BaTiO_3$ spherical nanoparticle covered with a $SrTiO_3$ shell and found Bloch points could form under certain conditions[35]. Recently, Gao et al. used the effective Hamiltonian approach to study the evolution of ultrathin $Pb(Zr_{0.4}Ti_{0.6})O_3$ film under an optical vortex beam and found the skyrmion state could be generated, accompanied with the emergence of a Bloch point[36]. However, observing Bloch points in these systems are extremely difficult. In the present work, we not only predicted the stabilization of polar Bloch points in tensile-strained ultrathin ferroelectric films by phase-field simulations, but

[1]Shenyang National Laboratory for Materials Science, Institute of Metal Research, Chinese Academy of Sciences, Wenhua Road 72, 110016 Shenyang, China. [2]Bay Area Center for Electron Microscopy, Songshan Lake Materials Laboratory, Dongguan 523808 Guangdong, China. [3]Quantum Science Center of Guangdong-HongKong-Macau Greater Bay Area, Shenzhen, China. [4]School of Materials Science and Engineering, Hunan University of Science and Technology, Xiangtan 411201, China. [5]School of Materials Science and Engineering, University of Science and Technology of China, Wenhua Road 72, Shenyang 110016, China. [6]Institute of Physics, Chinese Academy of Sciences, Beijing 100190, China. [7]State Key Lab of Advanced Processing and Recycling on Non-ferrous Metals, Lanzhou University of Technology, 730050 Lanzhou, China. [8]These authors contributed equally: Yu-Jia Wang, Yan-Peng Feng. ✉ e-mail: xlma@iphy.ac.cn

also observed their internal structures by aberration-corrected scanning transmission electron microscope (TEM). Moreover, we revealed the phenomenon of local steady-state negative capacitance around the Bloch point.

## Results

Based on our previous work of observing a polar meron lattice in an ultrathin PTO film grown on the orthorhombic (110)-oriented [or the pseudocubic (001)-oriented] SmScO₃ (SSO) substrate[30], we added a top electrode above the PTO film in the phase-field simulations to study the evolution of domain structures with the increase of the top electrode's thickness ($h_{te}$). As shown in Fig. 1a-c, the meron lattice gradually transfers into a Bloch point lattice with the increase of $h_{te}$. Figure 1d, e give the schematic diagrams of the transition from two typical merons [up-convergent (Fig. 1d) and down-divergent (Fig. 1e)] to two typical Bloch points. It is found that the original central $c$ domain in a meron shrinks close to the top surface and there emerged a new $c$ domain with the opposite polarization at the bottom surface. Bloch points locate in the central positions between the two $c$ domains, as marked by purple and yellow circles. During the transition, the in-plane polarization configuration seldom changes. Figure 1f, g shows the local polarizations around the two Bloch points. For one type of

Bloch points (purple), the in-plane (IP) polarization is convergent and the out-of-plane (OP) polarization is divergent, as shown in Fig. 1f. For the other type (yellow), the IP polarization is divergent and the OP polarization is convergent, as shown in Fig. 1g. In other words, among the three mutually perpendicular axes, the polarization vectors of the first type of Bloch points are convergent along two axes but divergent along the third axis (denoted as the CCD-type), while those of the second type of Bloch points are divergent along two axes but convergent along the third axis (denoted as the DDC-type). The polarization magnitudes are nearly zero at Bloch points, as revealed by the shrunk arrows at the centers of Fig. 1f, g. The topological charge of a polar Bloch point can be calculated by the following equation[37],

$$Q = \frac{1}{8\pi} \int dA_i \varepsilon_{ijk} \hat{\mathbf{p}}(\mathbf{r}) \cdot \left[ \partial_j \hat{\mathbf{p}}(\mathbf{r}) \times \partial_k \hat{\mathbf{p}}(\mathbf{r}) \right]. \tag{1}$$

where $\hat{\mathbf{p}}$ is the normalized polarization vector, $A_i$ is the surface of a small volume containing a Bloch point, $\varepsilon_{ijk}$ is the Levi-Civita symbol. The topological charges of the CCD- and DDC-type Bloch points are calculated as +1 (Fig. 1f) and −1 (Fig. 1g), respectively.

Figure 1h shows the statistical positions of Bloch points in the films with different thicknesses of the top electrode. It is interesting to

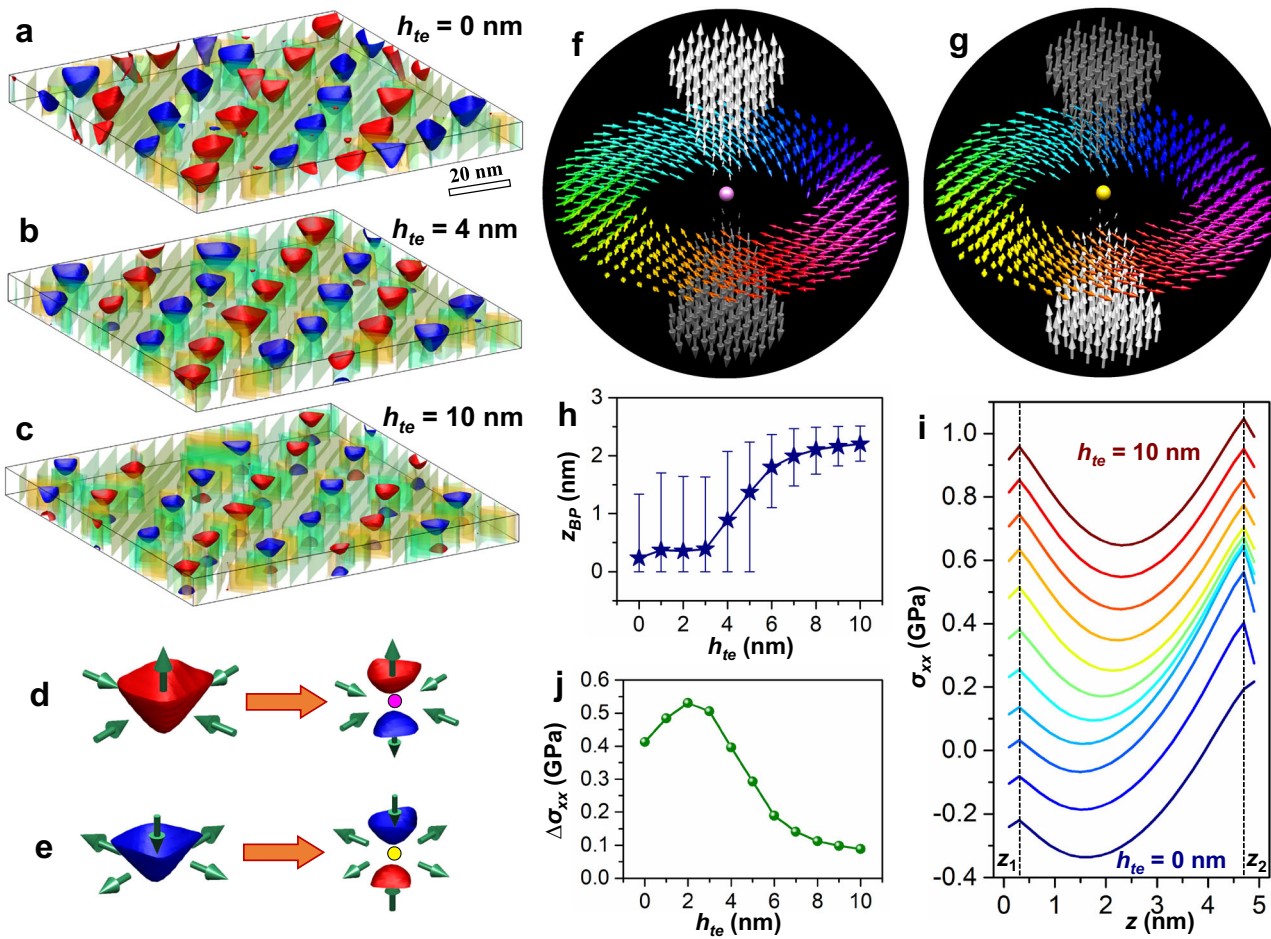

**Fig. 1 | The transition from merons to Bloch points and the local polarization configurations of polar Bloch points predicted by the phase-field simulations.** **a**–**c** The domain structures of a 5 nm PTO film capped with top electrodes of different thicknesses. The isosurfaces of $P_x$ ($P_y$, $P_z$) = ± 0.5 $P_s$ ($P_s$ = 0.76 C/m²) are plotted where $P_z$ = +0.5 $P_s$ (red) and $P_z$ = −0.5 $P_s$ (blue) are highlighted. For better visualization, $P_z$ = ± 0.7 $P_s$ are drawn in **a**. **d**, **e** The schematic diagrams of two types of merons transferring into two types of Bloch points. **f**, **g** The local polarization configurations of the two types of Bloch points. The purple and yellow circles

(balls) in **d**–**g** show the positions of Bloch points. **h** The averaged $z$ coordinates of Bloch points ($z_{BP}$) in the PTO film as the function of $h_{te}$. The up and down error bars show the maximal and minimal values. **i** The distribution of planar-averaged stress ($\sigma_{xx}$) in the PTO films with different $h_{te}$. $\sigma_{yy}$ is nearly the same as $\sigma_{xx}$. The profiles of $h_{te}$ = 1 - 10 nm are shifted upwards by 0.1 - 1 GPa, respectively. **j** The difference of $\sigma_{xx}$ near the top and bottom surfaces [$\Delta\sigma_{xx} = \sigma_{xx}(z_2) - \sigma_{xx}(z_1)$, where $z_1$ and $z_2$ are marked by two dashed lines in **i**] as the function of $h_{te}$.

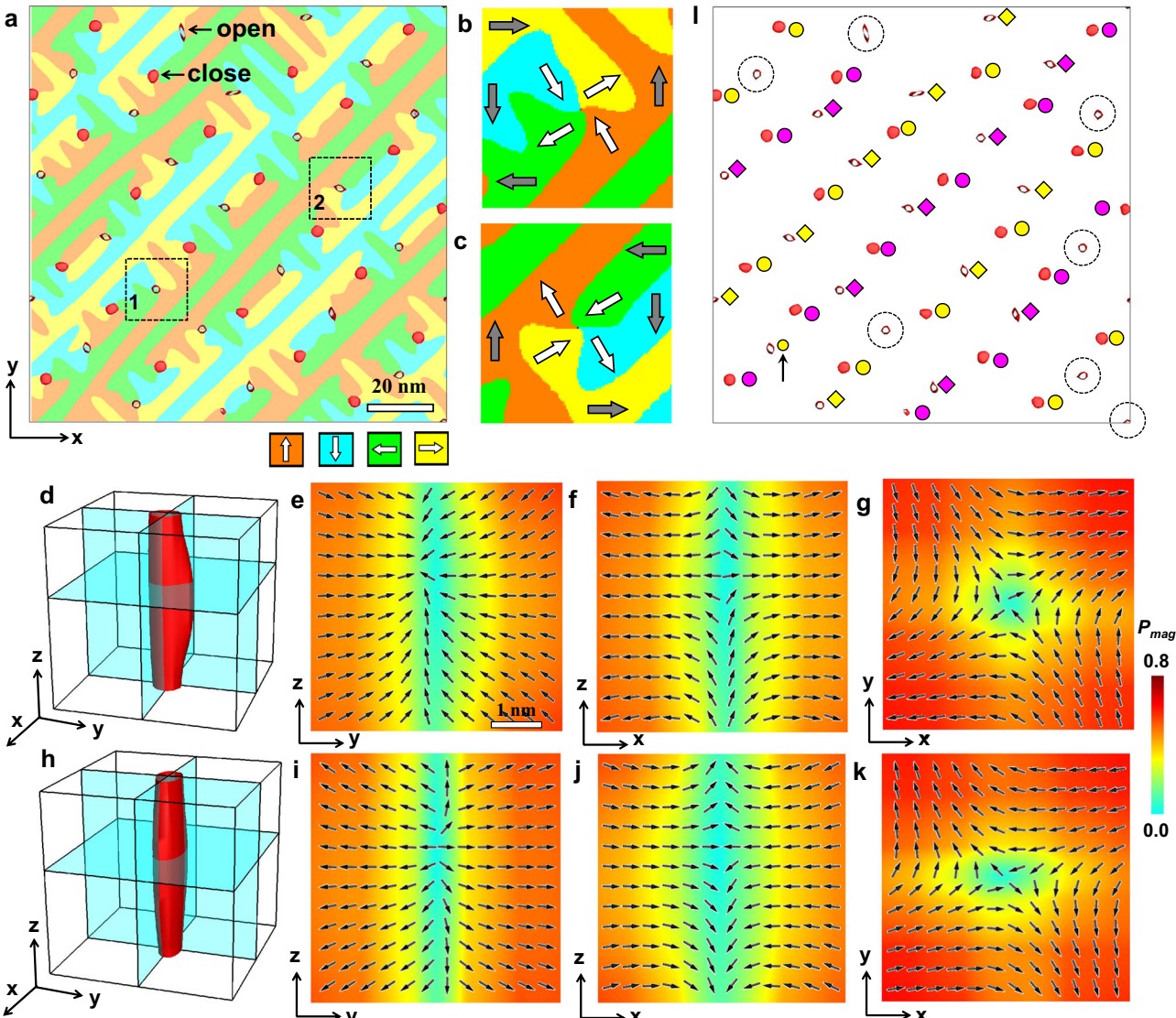

**Fig. 2 | Bloch points associated with antimerons predicted by the phase-field simulations. a** The horizontal slice of the central layer of the PTO film overlaid with the isosurfaces of $P_{mag} = 0.55\,P_s$ (red). The close isosurfaces are associated with BP-M's and most open isosurfaces BP-AM's. **b**, **c** The zoom-in slices of the two antimerons marked by dashed boxes in **a**. The white and gray arrows show the polarization directions near and away from the antimerons, respectively. **d**–**g** The isosurface of $P_{mag} = 0.3\,P_s$ (**d**) and the slices perpendicular to the $x$ (**e**), $y$ (**f**), and $z$ (**g**) axes for the antimeron in **b**. **h**–**k** The isosurface and the slices for the antimeron in **c**. The normalized polarization vectors are overlaid therein. **l** The isosurface of $P_{mag} = 0.55\,P_s$ (red) where the CCD- and DDC-type Bloch points are labeled by purple and yellow circles, respectively. The circles and diamonds represent BP-M's and BP-AM's, respectively. The black arrow labels the abnormal BP-AM and the dashed circles mark the antimerons not associated with Bloch points.

observe that when $h_{te}$ is 0 (corresponding to the case that there is no top electrode), there already exist some Bloch points and they are close to the bottom surface. With the increase of $h_{te}$, their positions shift to the center of the film. Figure 1i shows the profiles of the planar-averaged stress ($\sigma_{xx}$), which gradually changes from the asymmetric distribution to the nearly symmetric one as the increase of $h_{te}$. Figure 1j gives the difference of $\sigma_{xx}$ near the top and bottom surfaces. With the increase of $h_{te}$, the asymmetry first increases and then decreases. As explained in Supplementary Note 1, the introduction of the top electrode influences the mechanical driving force, which leads to the variation of the domain structure. These results indicate that it is the mechanical driving force that triggers the transition from merons to Bloch points.

From Fig. 1f, g, it is found that the Bloch points are associated with the regions of reduced polarization magnitude. Figure 2a gives the in-plane domain structure overlaid with the isosurfaces of $P_{mag} = 0.55\,P_s$.

There are two types of $P_{mag}$ isosurfaces: close and open. The close ones are associated with Bloch points in Fig. 1, while the open ones are related with antimerons. There are two types of antimerons: The polarizations of the first type are divergent/convergent along the approximate [110]/[1$\bar{1}$0] direction, as shown by white arrows in Fig. 2b. The second type is featured by convergent/divergent polarizations along the approximate [110]/[1$\bar{1}$0] direction, as shown by white arrows in Fig. 2c. The three-dimensional (3D) polarization distributions shown in Fig. 2d–k indicate that they are actually CCD- and DDC-type Bloch points. They are denoted as BP-AM, to be differentiated with those associated with merons (BP-M). Figure 2l marks the positions of all BP-M's and BP-AM's. It is found that BP-M's and BP-AM's on a straight line along the [110] direction are almost of the same type. However, one BP-AM does not obey this rule and some antimerons are not related with Bloch points, as marked by a black arrow and dashed circles and shown in detail in Supplementary Fig. 1. This random

behavior might be caused by the extremely small $P_z$ components around BP-AMs (<0.1 $P_s$), which can be easily influenced by the initial random noise.

According to the phase-field simulations, a sandwiched PTO thin film with SrRuO$_3$ (SRO) layers was grown on an orthorhombic (110)-oriented SSO substrate by pulsed laser deposition (PLD). The thickness of each PTO and SRO layer is about 5 nm. The upper and lower SRO layers were grown to form nearly short-circuit and meanwhile symmetric mechanical boundary condition (details of film deposition are provided in Methods). The surface topography of the trilayer film (Supplementary Fig. 2) displays that the film surface is rather smooth with a surface root-mean-square roughness of 153 pm. The atomic-resolved cross-sectional high-angle annular dark-field scanning transmission electron microscopy (HAADF-STEM) image (Supplementary Fig. 3) was acquired by an aberration-corrected scanning TEM, showing the thickness with 12–13 unit cells of each SRO and PTO layer. A typical low-magnification cross-sectional dark-field TEM image is shown as Supplementary Fig. 4a. It is noted that many dot contrast exists in the PTO layer, which might indicate the formation of the exotic domains here. A low-magnification planar-view dark-field TEM image (Supplementary Fig. 4b) exhibits the regular stripe $a_1/a_2$-like domains in films with the direction along the in-plane [110] or [1$\bar{1}$0] directions. Importantly, the stripe domain walls are not strictly straight. To further display the characteristic of the domain walls, a planar-view HAADF-STEM image acquired by the defocus mode was shown in Supplementary Fig. 4c, which features many dot contrast fluctuations at stipe domain walls, as marked by red arrows. Figure 3a shows the 3D sketch of a low-magnification planar-view TEM image

overlaid with the cross-sectional dark-field TEM images, displaying the overview of the SRO/PTO/SRO film. Figure 3b displays a superposition of reversed Ti-displacement vectors ($-\delta_{Ti}$) and the atomic-resolved HAADF-STEM image (The original one is provided as Supplementary Fig. 5). In Fig. 3b, the $-\delta_{Ti}$ vectors of PTO unit cells were marked by yellow arrows, which are consistent with the local spontaneous polarization directions[36]. To visualize the polarization configuration in the PTO film, four typical areas labeled as "I"–"IV" are magnified and shown as "1"–"4" in Fig. 3c, respectively. It is clearly seen that the areas "1" and "3" are two typical antivortex patterns, while the areas "2" and "4" exhibit two divergent polarization patterns. Importantly, the antivortex and divergent patterns are formed with alternating arrangement. Besides, atomic-resolved X-ray energy dispersive spectroscopy (EDS) was acquired to identify the element distribution in the film, as shown in Fig. 3d. It is seen that both interfaces are very sharp and there is no indication of inter-diffusion.

Similarly, the $-\delta_{Ti}$ vector map based on the atomic-resolved planar-view HAADF-STEM image (Supplementary Fig. 6) was extracted and shown in Fig. 3e, where the arrows with different colors denote the polarization directions of PTO unit cells. The high-resolution $-\delta_{Ti}$ vector map was displayed in Supplementary Fig. 7. The ordered stripe domains were identified in this PTO film, which is similar to $a_1/a_2$ domains in tetragonal ferroelectric films. What is different, however, is that many "head-to-head" (h-t-h) and "tail-to-tail" (t-t-t) DWs are formed in this PTO film, as marked by black and white arrows, respectively. Importantly, at each h-t-h DW (labeled by black solid lines in Fig. 3e), many convergent polarization patterns (labeled by black solid circles in Fig. 3e) exist, approximately forming one-dimensional array. It is

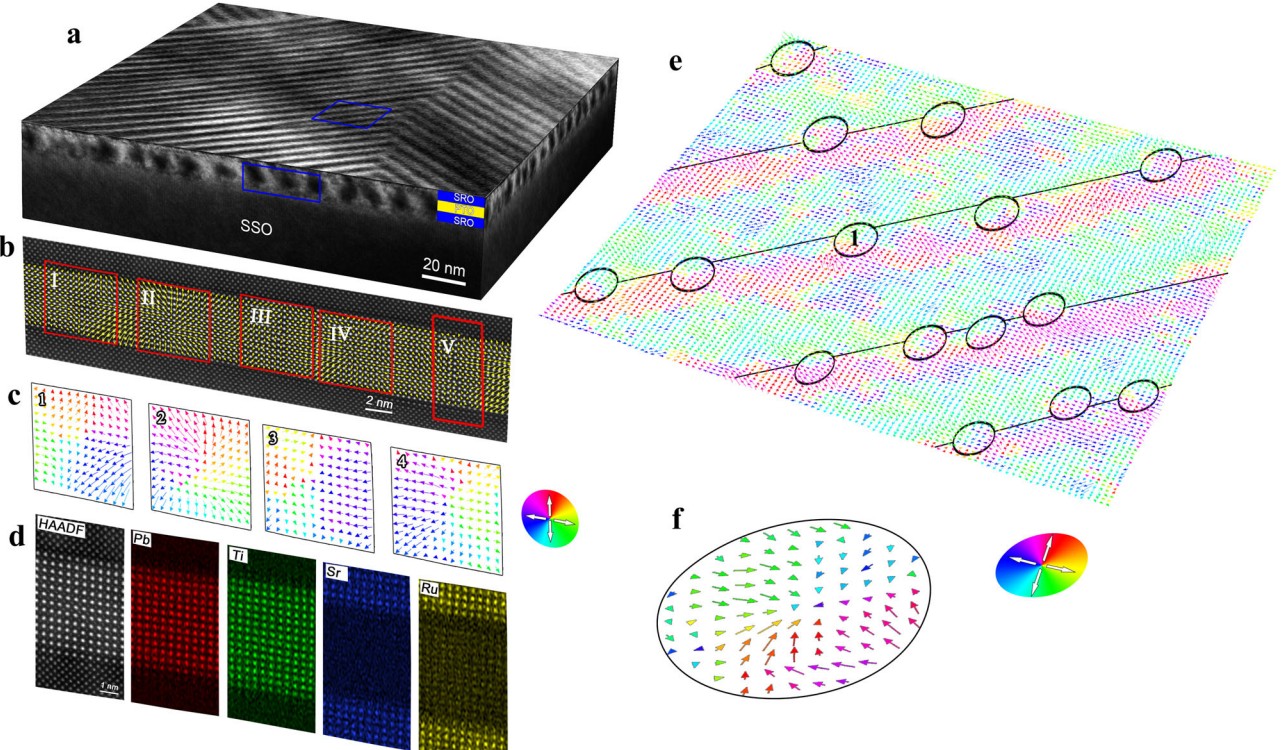

**Fig. 3 | Observation of polar Bloch points in a trilayer SRO/PTO/SRO thin film grown on the SSO substrate. a** The 3D sketch of the low-magnification planar-view TEM image overlaid with the cross-sectional dark-field TEM images, displaying the overview of the film. **b** Superposition of reversed Ti-displacement vectors ($-\delta_{Ti}$) and the atomic-resolved HAADF-STEM image. The $-\delta_{Ti}$ vectors of PTO unit cells were shown as yellow arrows. **c** Magnified $-\delta_{Ti}$ vector maps showing four typical polarization patterns marked as "1"–"4" in the thin film corresponding to four red boxes labeled as "I"–"IV" in **b**, respectively. **d** Atomic-resolved HAADF-STEM image

and EDS mapping of the region in the red box labeled as "V" in **b** showing the sharp interfaces of SRO/PTO and PTO/SRO. **e** The $-\delta_{Ti}$ vector map based on the atomic-resolved planar-view HAADF-STEM image, showing the convergent polarization patterns (black circles) at head-to-head domain walls (black lines). The arrows with different colors denote the polarization directions of PTO unit cells. **f** The magnified $-\delta_{Ti}$ vector map of a typical convergent polarization pattern corresponding to the area in **e**.

noted that the stripe domains are arranged periodically with the period of ~10 nm, as shown in Supplementary Fig. 4. Thus, the periodical arrangement of convergent polarization pattern arrays at h-t-h DWs are formed, as shown in Fig. 3e. The periodicity is consistent of the width of stripe $a_1/a_2$ domains. A typical area in Fig. 3e is selected and its zoom-in polarization map is shown in Fig. 3f, showing a convergent polarization configuration. On the other hand, many divergent polarization patterns are observed at t-t-t DWs, which also form one-dimensional arrays with periodical arrangement (Supplementary Fig. 8).

From Figs. 1 and 2, it is found that all cross-sectional slices of a BP-M are in the antivortex pattern and only the BP-AM could display convergent or divergent patterns in its cross-sectional slice. As mentioned before, the $P_z$ components around a BP-AM is generally smaller than $0.1 P_s$. However, the polarization vectors around the BP-AM's in experiments are comparable to the bulk polarization. We propose a hypothesis that there exists a local electric field pointing from PTO to SRO at the SRO/PTO interfaces, possibly due to their work function difference[38]. To consider this effect, an electric field divergent in the $z$ direction was applied to the PTO layer in the simulation model. Supplementary Fig. 9 gives the schematic of the electric field distribution and the resulted domain structure. It is found that the divergent electric field will enlarge the $c$ domain pairs of the CCD-type Bloch points, while shrink those of the DDC-type Bloch points, as shown in Supplementary Fig. 9c, d. Another change is that the polarization become divergent in the $z$ direction at all antimerons. As a result, all antimerons transfer into the DDC-type Bloch points, as shown in Supplementary Figs. 9e, f and 10. The comparison of the experimental and simulation cross-sectional polarization configurations is shown in Supplementary Fig 11. It is found that both the antivortex and convergent patterns are reproduced by the phase-field simulations, supporting the experimental observation. It is also noted that the h-t-h and the t-t-t DWs appear in pairs. Thus we also did simulations considering the distance between DWs. As shown in Supplementary Fig. 12, as the decrease of the distance between the h-t-h DW and the t-t-t DW, some Bloch points annihilate with antimerons. The number of Bloch points associated with merons decreases from 24 to 21 and 16 when the distance ratio $d_1/d_0$ decreases from 0.5 to 0.4 and 0.3. This is also consistent with the experimental observation that the distribution of Bloch points on one DW is actually not uniform.

From both the simulation and experimental results, it is explicit that there exists a region of zero polarization at the polar Bloch point, just like the cases of vortex core[39] and skyrmion periphery[40]. As shown in Supplementary Fig. 13, the free energy of a typical ferroelectrics is in the shape of a double well. In the state of zero polarization, the curvature of the free energy with respect to the polarization, which is the reciprocal dielectric constant $\varepsilon^{-1}$, is negative. Thus, we calculated the relative dielectric constant of this film to figure out the physical behaviors resultant from Bloch points. Figure 4a shows the isosurfaces of $P_{mag} = 0.55 P_s$ (red) and $\varepsilon_{zz} = -5000$ (green). It is found that the two isosurfaces are closely related and the regions of negative capacitance associated with BP-M's are much larger than those associated with BP-AM's. Thus, we mainly discuss the negative capacitance associated with BP-M's. Figure 4b shows the local polarization around a DDC-type Bloch point. The IP polarizations show a divergent vortex pattern (Fig. 4c), while the vertical cross-section exhibits an antivortex pattern (Fig. 4d). The corresponding electric field distributions are drawn in Fig. 4e, f. The IP electric field shows a convergent distribution around the Bloch point, as shown in Fig. 4e. In the vertical slice (Fig. 4f), the electric field shows a peculiar behavior, which is antiparallel to the polarization near the Bloch point but parallel to the polarization away from the Bloch point, as also shown in the line profiles in Fig. 4g. Correspondingly, $\varepsilon_{zz}$ exhibits the regions of positive (red) and negative (blue) capacitances, as shown in Fig. 4h. Supplementary Fig. 14 shows

the polarization and electric field distribution around a CCD-type Bloch point. Similar behaviors can be observed.

To further understand the antiparallel behavior, the bound charge around the Bloch points was analyzed. By comparing the isosurfaces of bound charge (Supplementary Fig. 15b) with those of $P_z$ (Supplementary Fig. 15e) for the CCD-type Bloch point marked by "1" in Supplementary Fig. 15a, it is found that there exists negative bound charge just at the Bloch point. However, there also exist positive ones above and under this Bloch point. As a result, there exists a convergent electric field around this Bloch point. That is the reason why the electric field is opposite to the polarization around the Bloch point. Similarly, there exist a positive bound charge region and two satellite negative bound charge regions at the DDC-type Bloch point marked by "2" in Supplementary Fig. 15a and the local divergent electric field is formed, as shown in Supplementary Fig. 15c. These particular bound charge distributions result from the delicate balance between the IP and OP polarization distributions. Supplementary Fig. 15g show the profiles of IP ($\sigma_{IP} = -\partial P_x/\partial x - \partial P_y/\partial y$), OP ($\sigma_{OP} = -\partial P_z/\partial z$) and total ($\sigma_{tot} = \sigma_{IP} + \sigma_{OP}$) bound charges along a vertical line through the CCD-type Bloch point. It is found that $\sigma_{IP}$ is positive due to the convergent IP polarization configuration and $\sigma_{OP}$ is negative due to the divergent OP polarization configuration. Their common effect results in a positive-negative-positive distribution of the total bound charge $\sigma_{tot}$. Supplementary Fig. 15h show these profiles for the DDC-type Bloch point, which are just opposite to those in Supplementary Fig. 15g. From this bound charge analysis, it is understood that the negative capacitance originate from the delicate balance between the IP and OP polarization configurations around the Bloch point.

The present study indicates that by engineering mechanical and electric boundary conditions in ferroelectric films, topological polar Bloch points could be produced. The resultant negative capacitance around a polar Bloch point further strengthen the relationship between polar topological structures and negative capacitance. This work not only enlarges the assembly of topological ferroelectric domain structures, but also extends the boundary of ferroelectric negative capacitance. More works should be done to explore the dynamical properties of polar Bloch points, including experimental verification of negative capacitance in this system. Moreover, it is also worthwhile to figure out whether Bloch points could emerge in other ferroelectric systems.

## Methods

### Phase-field modeling

The topological structures in PTO films was studied by phase-field simulations. The adopted order parameters are the three components of the spontaneous polarization. The system's free energy is the functional of the spontaneous polarization and composed of the bulk, gradient, elastic and electrostatic ones:

$$F = \int_V \left[ f_{bulk}(P_i) + f_{grad}(P_{i,j}) + f_{elas}(P_i, \varepsilon_{kl}) + f_{elec}(P_i, E_i) \right] dV \quad (2)$$

The first term is the bulk energy density:

$$\begin{aligned} f_{bulk} = & \alpha_1(P_1^2 + P_2^2 + P_3^2) + \alpha_{11}(P_1^4 + P_2^4 + P_3^4) \\ & + \alpha_{12}(P_1^2 P_2^2 + P_2^2 P_3^2 + P_1^2 P_3^2) + \alpha_{111}(P_1^6 + P_2^6 + P_3^6) \\ & + \alpha_{112}[P_1^4(P_2^2 + P_3^2) + P_2^4(P_1^2 + P_3^2) + P_3^4(P_1^2 + P_2^2)] + \alpha_{123}P_1^2 P_2^2 P_3^2 \end{aligned} \quad (3)$$

The second term is the gradient energy density:

$$\begin{aligned} f_{grad} = & \tfrac{1}{2} G_{11}(P_{1,1}^2 + P_{2,2}^2 + P_{3,3}^2) + G_{12}(P_{1,1}P_{2,2} + P_{2,2}P_{3,3} + P_{1,1}P_{3,3}) \\ & + \tfrac{1}{2} G_{44}[(P_{1,2} + P_{2,1})^2 + (P_{2,3} + P_{3,2})^2 + (P_{3,1} + P_{1,3})^2] \end{aligned} \quad (4)$$

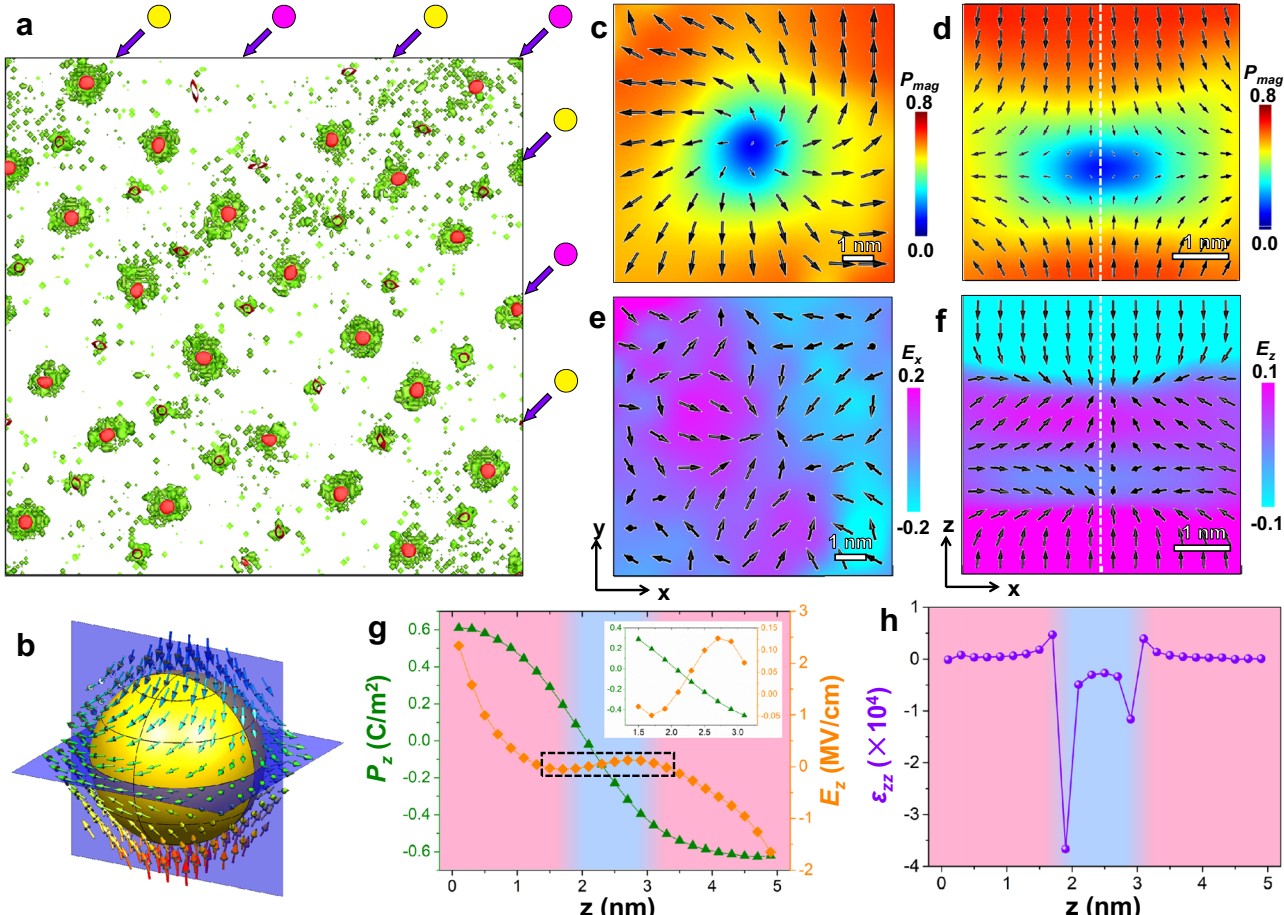

**Fig. 4 | Local negative capacitance around polar Bloch points revealed by the phase-field simulations. a** The isosurface of $\varepsilon_{zz} = -5000$ (green) overlaid with that of $P_{mag} = 0.55\,P_s$ (red). The chains of two types of Bloch points are marked by purple and yellow circles. **b** The local polarization around a DDC-type Bloch point with the $xy$ and $xz$ cross-sections passing through the Bloch point. **c, d** The mappings of the polarization magnitude overlaid with the polarization vectors of the horizontal (**c**) and vertical (**d**) slices of the Bloch point. **e, f** The corresponding mappings of the electric field overlaid with the normalized electric field vectors of the two slices in **c**, **d**. **g**, **h** The profiles of the polarization and electric field (**g**) and $\varepsilon_{zz}$ (**h**) along the dashed lines in **d** and **f**. The red and blue background colors represent the regions of positive and negative capacitances, respectively.

where $P_{i,j} = \frac{\partial P_i}{\partial x_j}$. The third term is the elastic energy density:

$$f_{elas} = \frac{1}{2}C_{ijkl}(\varepsilon_{ij} - \varepsilon_{ij}^0)(\varepsilon_{kl} - \varepsilon_{kl}^0) \qquad (5)$$

where $\varepsilon_{ij}$ is the total strain and $\varepsilon_{ij}^0$ is the spontaneous strain. Their difference is the elastic strain. The spontaneous strain $\varepsilon_{ij}^0 = Q_{ijkl}P_kP_l$ is related to the polarization by the electrostrictive coefficients $Q_{ijkl}$. The last term is the electrostatic energy density:

$$f_{elec} = -\frac{1}{2}\varepsilon_0\varepsilon_bE_i^2 - E_iP_i \qquad (6)$$

where $\varepsilon_0$ is the permittivity of vacuum and $\varepsilon_b$ is the background relative dielectric constant.

It is assumed that the equilibriums of the mechanical stress and the electrical field are much faster than the evolution of domain structures. Based on this assumption, the mechanical equilibrium equation, $\sigma_{ij,j} = \frac{\partial\sigma_{ij}}{\partial x_j} = 0$, and the Maxwell equation, $D_{i,i} = \frac{\partial D_i}{\partial x_i} = 0$, are solved firstly and the obtained driving forces are substituted into the time-dependent Ginzburg-Landau (TGDL) equation, $\frac{dP_i}{dt} = -L\frac{\delta F}{\delta P_i}$, to simulate the evolution of the domain structure. The backward Euler method is adopted in the solution of the TDGL equation. The

introduction of the top electrode influences the mechanical driving force, which will be explained in Supplementary Note 1.

For the simulations of Bloch points, the size of a typical simulation box is $128 \times 128 \times 100$, corresponding to the real-space size of $128 \times 128 \times 20$ nm³, which contains the down electrode (5 nm), the PTO film (5 nm) and the top electrode (10 nm). The top surface of the top electrode is in a traction-free state, while the bottom of the down electrode is fixed to the strain of SSO. The short-circuit electric boundary condition is applied to the top and bottom surfaces of the film. For the sake of simplicity, the elastic constants of the electrode layers were chosen as the same as those of PTO. All coefficients of PTO are adopted from a previous literature[41] and listed in Supplementary Table 1.

**The calculation of local dielectric constant**

The local dielectric constant ($\varepsilon$) is defined as the derivative of the local electric displacement (**D**) with respective to that of the local total electric field (**E**). It is a two-dimensional tensor in the form of a $3 \times 3$ matrix. In this paper, we focus on $\varepsilon_{zz} = \partial D_z/\partial E_z$. To calculate the local dielectric constant, a small electric field (400 kV/m) is applied across the film and the variance of the local total electric field ($\partial E_z$) and local electric displacement ($\partial D_z$) is calculated for each grid point of the simulation cell and $\varepsilon_{zz}$ could be calculated accordingly. The local electric displacement ($D_z$) is calculated as $D_z = \varepsilon_0\varepsilon_bE_z + P_z$.

## Film deposition details

The ferroelectric SRO/PTO/SRO trilayer films were deposited on orthorhombic (110)-oriented SSO single-crystal substrate by PLD, using the Coherent ComPex PRO 201 F KrF excimer laser ($\lambda = 248$ nm). The thicknesses of the PTO and SRO layers are controlled as about 5 nm. The sintered PTO target with 3 mol% Pb enrichment and stoichiometric SRO target were used to deposit PTO and SRO layers, respectively. Before film deposition, the SSO substrate was pre-heated to 800 °C and kept for 10 min to make the surface cleaning, and then cooled down to 700 °C for the deposition. The PTO and SRO targets were pre-sputtered for 5 min to clean the target surfaces, respectively. During the deposition of the SRO layer, an oxygen pressure of 7 Pa, a laser energy of 1.7 J cm$^{-2}$ and a repetition rate of 4 Hz were used, while an oxygen pressure of 10 Pa, a laser energy of 2 J cm$^{-2}$ and a repetition rate of 4 Hz were used when depositing the PTO layer. After the deposition, the sample was annealed at 700 °C for 5 min in an oxygen pressure of $3 \times 10^4$ Pa, and then cooled down to the room temperature with the cooling rate of 5 °C/min.

## Atomic force microscopy

Surface morphology measurements were performed by utilizing a commercial atomic force microscopy (Cypher-ES, Asylum Research, US). The contact mode is chosen in the measurements with a common silicon cantilever (Asylum Research, AC240TS-R3) whose tip radius and force constant are 7 nm and 2 N m$^{-1}$, respectively.

## HAADF-STEM imaging and determining the positions of atom columns

The traditional method with gluing, grinding, dimpling, ion milling was used to prepare cross-sectional TEM samples. A PIPS 691 (Gatan) was used for final ion milling. At first, the voltage of 4.5 kV and the incident angle of 7° were used for the ion milling. Then, the angles were gradually reduced to 5° and the final voltage was reduced to 0.5 kV to clean the amorphous on the surfaces of samples. The similar method was used to prepare planar-view TEM samples, which were ion-milled only from the substrate side.

All of cross-sectional and planar-view HAADF-STEM images were acquired by an aberration-corrected scanning TEM (Titan Themis 60–300 X-FEG microscope (Thermo Fisher Scientific)) with double aberration (Cs) correctors from CEOS and a monochromator, which was operated at 300 kV. The atomic-resolved HAADF-STEM images for acquiring the polarization map are recorded by STEM Drift Corrected Frame Integration (DCFI). The DCFI technique records successive original STEM images and integrates a high-quality STEM image via calculating and correcting the drift from the cross correlation[42–44]. Each high-resolution HAADF-STEM image was acquired by adding up 20 original images with the dwell time of 200 ns, which was performed by using the Velox software (Thermo Fisher Scientific). The positions of atom columns in HAADF-STEM images were determined on the basis of the two-dimensional Gaussian fitting, which was carried out by using the Matlab software[45,46].

## Reporting summary

Further information on research design is available in the Nature Portfolio Reporting Summary linked to this article.

## Data availability

The data sets generated and analyzed during the current study are available from the corresponding author on reasonable request. Source data for Figs. 1–4 are provided with the paper.

## Code availability

The codes for the phase-field simulation in the current study are available from the corresponding author on reasonable request.

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

## Acknowledgements

This work is supported by the National Natural Science Foundation of China (no. 52122101, no. 51971223), Guangdong Provincial Quantum Science Strategic Initiative (no. GDZX2202001, GDZX2302001) and Shenyang National Laboratory for Materials Science (L2022R04, L2021F06). Y.J.W. acknowledges the Youth Innovation Promotion Association CAS (no. 2021187). Y.P.F. acknowledges the China National Postdoctoral Program for Innovative Talents (no. BX2021348), the Guangdong Basic and Applied Basic Research Foundation (no. 2021A1515110064, no. 2023A1515011058). Y.L.T. acknowledges the Scientific Instrument Developing Project of CAS (YJKYYQ20200066) and the Youth Innovation Promotion Association of CAS (Y202048).

## Author contributions

X.L.M. and Y.L.Z. conceived the project on architecture of quantum materials modulated by ferroelectric polarizations; X.L.M., Y.L.T., Y.L.Z., and Y.P.F. designed the sample structure and subsequent experiments. Y.P.F. performed the thin-film growth and STEM observations. Y.J.W. performed phase-field simulations and carried out digital analysis of the STEM data; Y.C., M.J.Z., and W.R.G. participated in the thin-film growth and STEM observations. All authors participated in discussion and interpretation of the data. Correspondence and requests for materials should be addressed to X.L.M.

## Competing interests

The authors declare no competing interests.
