## [Peer Review File · Nature Communications]

Polar Bloch points in strained ferroelectric filmsREVIEWER COMMENTS

Reviewer #1 (Remarks to the Author):

This work titled "Polar Bloch points in strained ferroelectric films" done by Y. J. Wang et al. reported the observation of polar Bloch points, in tensile-strained ultrathin ferroelectric PbTiO₃ films. Detailed phase-field simulations and TEM results were acquired, and the manuscript was well organized. Their result contains sufficient novelty to the field and is worthy to be published in Nature Communications. However, the following concerns need to be adequately addressed before I recommend publication.

1. Figure 3 is essential evidence for the existence of polar Bloch points. However, current Figures a (planar view), b, c (cross-sectional view), and e (polarization of planar view) are too small to identify. I suggest adjusting the layout of Figure 3 for a more clear view.
2. Could the author further provide DF-TEM images of the cross-section view and low-magnification HAADF/MAADF images of the planar view? To a certain extent, should the polarization patterns of the cross-section in Figure 3b correspond to the contrast pattern of the dark field image? Do the polarization patterns in Figure 3e correspond to the contrasts in the HAADF/MAADF-STEM image?
3. In supplementary Fig. 8, "convergent polarization patterns" should be "divergent...", in my point of view. In supplementary Fig. 10, "three" should be "four".
4. According to the authors' statement on page 7, both convergent polarization patterns at h-t-h DWs and divergent polarization patterns at t-t-t DWs form lattices. Is there any conclusion that the polar Bloch points favor constructing a periodic array? The phase field simulation results are present sort of periodic. The author is advised to provide some evidence if it is not too harsh. Some discussions about periodicity may be necessary, as before [A. K. Yadav et al., Nature, 530, 198–201 (2016)], [S. Das et al., Nature, 568, 368–372 (2019)] and [F. H. Gong et al., Science Advances, 7, 28 (2021)],
5. From the experimental planar views, the convergent polarization patterns at h-t-h DWs and divergent polarization patterns at t-t-t DWs always appear in pairs, as shown in Supplementary Fig. 7 and 8. From the cross-sectional views, a pair of anti-vortex and divergent polarization patterns are close to each other, as shown in Supplementary Fig. 10. But in the simulations, they are strictly independent and arranged periodically. Is there any explanation for this?
6. Bloch point is a singular point around which the vectors orient in nearly all directions. From the cross-sectional view, the polarization around the Bloch point is nearly uniform in all directions. However, in the planar view (such as Supplementary Fig. 8), many of the marked configurations around which the polarization is incomplete. Such as the marks on the fourth and fifth dashed lines, have almost no polarization pointing towards the upper right direction. In this case, determining the polar Bloch points and discussing their distribution just from the polar map may not be accurate. In addition, I saw some typical anti-vortex configurations at h-t-h DWs. Is it reasonable in the planar view of Bloch points system?

Reviewer #2 (Remarks to the Author):

I have had a chance to read this paper. It is a nice paper, but I would like the authors to clarify a few things :

1. The authors need to give a more convincing answer as to WHY such Bloch points form in a uniform PTO layer. It has a metallic SRO bottom electrode as as such, at least on one contact electrode they should be efficient screening of the surface charge. So, what is the need to form such exotic Bloch points? Is it possible that their SRO layer does not have the bulk metallicity (or carrier concentration) that results in such exotic domain formation? Some additional information on the metallicity of SRO is needed.

The inference of negative capacitance at the Bloch points is indeed interesting and I would have expected this to happen.

Overall, this is a good piece of work and can be published after the question above is answered.

Reviewer #3 (Remarks to the Author):

Please see the attachment Report_436889.pdf.

Wang and Feng et al. reported the emergence of one topological defect—polar Bloch point in strained ferroelectric PbTiO_3 . The study is supported by both numerical simulation and experimental observation. The work is interesting and novel in that ferroelectric Bloch point has only been predicted once by theory (Ref. 35 cited by authors). Therefore, this work has the potential to open a new research area about exploring Bloch points in ferroelectrics.

Nevertheless, after carefully reviewing this manuscript, certain sections appear either scientifically unsound or lacking in clarity. Moreover, additional evidence is necessary to substantiate the claim. With my background being a computational/theoretical physicist, I would like to note that Figs. 3(a)(b)(d) in the main manuscript and Figs.S2-S6 are outside the scope of my expertise. Below are my critiques:

1. For magnetic Bloch point, owing to its topology, it is usually associated with the creation or annihilation of a skyrmion (for example, see Fig. 18 in Review of Modern Physics 89, 025006 (2017)). Nevertheless, I didn't see any formation/deformation of polar skyrmion in this study. The authors should give remarks about this point.
2. In simulation, the authors discovered the Bloch points by leaving an electrode on top of the thin film. The authors should give an illustration about how this set up looks like in experiment in Fig 1. Also, the authors explored positions of Bloch points by varying the electrode thickness; however, I didn't see what variables electrode thickness correspond to in free energy. This must be mentioned in the method section.
3. From Line 79 to Line 84: the authors quickly drew a conclusion that a mechanical force/stress drives the topological transition from meron to Bloch points. It's totally elusive why the mechanical force should be related to electrode thickness and be the decisive factor of topological transition. To me the thickness of electrode is not related to strain at all. Also, I don't understand what a zero thickness of electrode physically corresponds to.
4. Figure 2 must be improved. The current version is very confusing. In Fig. 2(a), what does the color denote—polarization magnitude or direction? No colorbar is presented there. I don't understand what those stripes composed of 5 colors stand for. What do red circles represent? I know they are isosurfaces but are they meron-Bloch points? In Fig. 2(l), I cannot differentiate Meron- and Antimeron- Bloch points, though authors claim they correspond to large and small circles but it's very hard to tell from the figure. Are Meron-Bloch points red circles, and Antimeron-Bloch points yellow and purple circles? Are red circles in Fig. 2(l) the same as red circles in Fig. 2(a)?
5. The authors explained why no zero-polarization is observed in experiments by proposing a hypothesis that there is a quadratic-like electric field divergent along z direction. Though by simulation, it can give resembling polarization patterns to experiments (Fig. S10), I cannot relate the formation of such electric field pattern (Fig. 9b) with any physical charge distribution resulting from work function difference. In my opinion, such explanation does not correspond to any realistic scenario and is not physical.
6. Considering there is no zero-polarization region observed in experiments, I am wondering if there is any other experimental evidence to support the emergence of Bloch points, in addition to Fig. 3(b). For example, what would be the other orthogonal cross-sectional view of Section (I)-(IV)? Does that agree with calculation in Figs. 1 and 2?
7. On Line 177: "all antimerons transform into DDC-type Bloch points". Is it CCD instead? Fig. 3c (and Line 150) shows they are divergent, while Line 180 mentions it as "convergent". This is

really confusing. I also don't understand conversions of DDC to CCD Bloch points shown in Fig. S9(e)(f); I didn't see a drastic change of the numbers of yellow (or purple) circles, while the authors claimed one type has transformed to the other.

Other minor issues:

1. On Line 103, 104: the authors mention "close" and "open" isosurfaces. I assume the authors refer to the geometry shown in Figs. 1(d)(e) and Figs. 2(d)(h); the authors should give clear indications about these terminologies.
2. On Line 130: I assume the PTO is grown along [001] direction. The authors should specify the growth direction.
3. In Fig. 3e, the convergent polarization patterns don't form in a periodic pattern; I don't understand why the authors would like to call it an "approximated lattice."
4. The logic from local zero polarization to negative capacitance is not explicit; the authors should make the transition smoother there.
5. How to calculate local D in phase-field simulation? Is it to calculate local P and E and then get D from $D = \epsilon_0 E + P$? This point should be made clear in the method section.
6. Figures similar to Figs. 4(b)-(f) have been presented in the main manuscript; I would suggest swapping them with Figs. S12(b)(c)(e)(f)(g)(h) in the main manuscript.

Response to Reviewers' Comments:

Ref: NCOMMS-23-27559

Title: Polar Bloch points in strained ferroelectric films

02 February 2024

We appreciate the general cognition by the reviewers that “Their result contains sufficient novelty to the field and is worthy to be published in Nature Communications”, “this work has the potential to open a new research area about exploring Bloch points in ferroelectrics” and that “The inference of negative capacitance at the Bloch points is indeed interesting and I would have expected this to happen”. Nevertheless, the reviewers also raised some concerns regarding some experimental and simulation details. We fully understand the concerns by the reviewers, and we have addressed each of these concerns point-to-point in the following. Key revisions are highlighted in **RED** in the revised manuscript.

Reviewer #1

Comment #1

Figure 3 is essential evidence for the existence of polar Bloch points. However, current Figures a (planar view), b, c (cross-sectional view), and e (polarization of planar view) are too small to identify. I suggest adjusting the layout of Figure 3 for a more clear view.

Response

We thank the reviewer for the suggestion to modify Figure 3. We have enlarged Figures 3a, 3b, 3c and 3e to make it more clear for the identification of polar Bloch points. We have also replaced the cross-sectional and planar-view images in Figure 3a with more clear images. The modified Figure 3 is also presented here as Figure R1.

Fig. R1. Observation of polar Bloch points in trilayer SRO/PTO/SRO thin film grown on the SSO substrate. (a) The 3D sketch of low-magnification planar-view TEM image overlaid with the cross-sectional **dark-field TEM** images, displaying the overview of the film. (b) Superposition of reversed Ti-displacement vectors ($-\delta_{Ti}$) and atomic-resolved HAADF-STEM image. The $-\delta_{Ti}$ vectors of PTO unit cells were shown as yellow arrows. (c) Magnified $-\delta_{Ti}$ vector maps showing four typical polarization patterns marked as “1”-“4” in the thin film corresponding to four red boxes labeled as “I”-“IV” in (b), respectively. (d) Atomic-resolved HAADF-STEM image and EDS mapping of **the region in the red box labeled as “V” in (b)** showing the sharp interfaces of SSO/PTO and PTO/SSO(001)_{pc}. (e) The $-\delta_{Ti}$ vector map based on atomic-resolved planar-view HAADF-STEM image, showing the convergent polarization patterns (black circles) at head-to-head domain walls (black lines). The arrows with different colors denote the polarization directions of PTO unit cells. (f) The magnified $-\delta_{Ti}$ vector map of a typical convergent polarization pattern corresponding to the area in (e).

Comment #2

Could the author further provide DF-TEM images of the cross-section view and low-magnification HAADF/MAADF images of the planar view? To a certain extent, should the polarization patterns of the cross-section in Figure 3b correspond to the contrast pattern of the dark field image? Do the polarization patterns in Figure 3e correspond to the contrasts in the HAADF/MAADF-STEM image?

Response

We appreciate the valuable comment. We acquired the cross-sectional dark-field TEM image and the planar-view HAADF-STEM image, which were updated as Supplementary Fig. 4a and 4c,

respectively (also provided here as Fig. R2). In Fig. R2a, it is noted that many dot contrast exists in the PTO layer, which might indicate the formation of the exotic domains here. In Fig. R2c, it is obviously seen that the stripe domain walls are not strictly straight with many dot contrast fluctuations at stripe domain walls. These characteristics indicate the formation of the exotic domains in PTO films.

Changes

To make the statement clearly, we modified the main text as seen in RED on Page 7 “A typical low-magnification cross-sectional dark-field transmission electron microscopy (TEM) image is shown as Supplementary Fig. 4a. It is noted that many dot contrast exists in PTO layer, which might indicate the formation of the exotic domains here. A low-magnification planar-view dark-field TEM image (Supplementary Fig. 4b) exhibits the regular stripe a_1/a_2 -like domains in films with the direction along the in-plane $[110]$ or $[1\bar{1}0]$ directions. Importantly, the stripe domain walls are not strictly straight. To further display the characteristic of the domain walls, a planar-view HAADF-STEM image acquired by defocus mode was shown in Supplementary Fig. 4c, which features many dot contrast fluctuations at stripe domain walls, as marked by red arrows. Fig. 3a shows the 3D sketch of a low-magnification planar-view TEM image overlaid with the cross-sectional dark-field TEM images, displaying the overview of the SRO/PTO/SRO film”.

We also added cross-sectional dark-field TEM image and the planar-view HAADF-STEM image as Supplementary Fig. 4a and 4c, respectively in revised Supplementary Materials and used them to constitute the 3D sketch of Fig. 3a.

Fig. R2 (a) A low-magnification cross-sectional TEM image displays dot contrasts in the PTO layer. (b) A low-magnification planar-view dark-field TEM image exhibits the regular stripe a_1/a_2 -like domains. The inset is the magnified image corresponding to the area marked by a white box in (b), which indicates that the width of these stripe a_1/a_2 -like domains is about 10 nm. (c) A planar-view

HAADF-STEM image acquired by the defocus mode shows dot contrast fluctuations at stipe domain walls.

Comment #3

In supplementary Fig. 8, “convergent polarization patterns” should be “divergent...”, in my point of view. In supplementary Fig. 10, “three” should be “four”.

Response

Thank you greatly for carefully checking our manuscript. We have modified the two places accordingly.

Comment #4

According to the authors' statement on page 7, both convergent polarization patterns at h-t-h DWs and divergent polarization patterns at t-t-t DWs form lattices. Is there any conclusion that the polar Bloch points favor constructing a periodic array? The phase field simulation results are present sort of periodic. The author is advised to provide some evidence if it is not too harsh. Some discussions about periodicity may be necessary, as before [A. K. Yadav et al., Nature, 530, 198–201 (2016)], [S. Das et al., Nature, 568, 368–372 (2019)] and [F. H. Gong et al., Science Advances, 7, 28 (2021)],

Response

We fully understand the concerns raised by the reviewer. The convergent polarization patterns are arranged along each h-t-h DW, forming a one-dimensional convergent polarization pattern array, as shown in Fig. 3e. Similarly, the divergent polarization patterns are arranged along each t-t-t DW, forming a one-dimensional divergent polarization pattern array. Indeed, the interspacing between convergent (divergent) polarization patterns is not uniform. However, the h-t-h DWs and t-t-t DWs are arranged periodically. The periodicity is consistent of the width of stripe a_1/a_2 domains, which is about 10 nm, as shown in the inset of Fig. R2b. Thus, the convergent polarization pattern arrays and divergent polarization pattern arrays are alternately arranged with periodicity of 10 nm.

Change

To make the statement clearly, we have modified the main text as seen in RED on Page 8, as “Importantly, at each h-t-h DW (labeled by black solid lines in Fig. 3e), many convergent polarization patterns (labeled by black solid circles in Fig. 3e) exist, approximately forming one-dimensional array. It is noted that the stripe domains are arranged periodically with the period of about 10 nm, as shown in Supplementary Fig. 4. Thus, the periodical arrangement of convergent polarization pattern arrays at h-t-h DWs are formed, as shown in Fig. 3e. The periodicity is consistent of the width of stripe a_1/a_2 domains. A typical area in Fig. 3e is selected and its zoom-in polarization map is shown in Fig. 3f, showing a convergent polarization configuration. On the other hand, many divergent polarization patterns are observed at t-t-t DWs, which also form one-dimensional arrays with periodical arrangement (Supplementary Fig. 8).”

Comment #5

From the experimental planar views, the convergent polarization patterns at h-t-h DWs and divergent

polarization patterns at t-t-t DWs always appear in pairs, as shown in Supplementary Fig. 7 and 8. From the cross-sectional views, a pair of anti-vortex and divergent polarization patterns are close to each other, as shown in Supplementary Fig. 10. But in the simulations, they are strictly independent and arranged periodically. Is there any explanation for this?

Response

Thank you for the valuable comment. We have done more simulations, considering the distance between DWs. As shown in Fig. R3, as the decrease of the distance between the h-t-h DW and the t-t-t DW, some Bloch points annihilate with antimerons. The number of Bloch points associated with merons decreases from 24 to 21 and 16 when the distance ratio d_1/d_0 decreases from 0.5 to 0.4 and 0.3. This is also consistent with the experimental observation that the distribution of Bloch points along one array is actually not uniform.

Fig. R3. The domain structures from the models with different initial distances between DWs. d_1 is the distance between the h-t-h and t-t-t DWs and d_0 is the distance between two h-t-h DWs. (a) $d_1/d_0 = 0.4$. (b) $d_1/d_0 = 0.3$. The results of $d_1/d_0 = 0.5$ is shown in Fig. 2(a). The number of Bloch points associated with merons decreases from 24 to 21 and 16 when the distance ratio d_1/d_0 decreases from 0.5 to 0.4 and 0.3.

Change

We have added Fig. R3 as Supplementary Fig. 12 in the revised supplementary materials and added several sentences (It is also noted that the h-t-h and the t-t-t DWs appear in pairs. Thus we also did simulations considering the distance between DWs. As shown in Supplementary Fig. 12, as the decrease of the distance between the h-t-h DW and the t-t-t DW, some Bloch points annihilate with antimerons. The number of Bloch points associated with merons decreases from 24 to 21 and 16 when the distance ratio d_1/d_0 decreases from 0.5 to 0.4 and 0.3. This is also consistent with the experimental observation that the distribution of Bloch points on one DW is actually not uniform.) on Page 8-9 in

the revised manuscript.

Comment #6

Bloch point is a singular point around which the vectors orient in nearly all directions. From the cross-sectional view, the polarization around the Bloch point is nearly uniform in all directions. However, in the planar view (such as Supplementary Fig. 8), many of the marked configurations around which the polarization is incomplete. Such as the marks on the fourth and fifth dashed lines, have almost no polarization pointing towards the upper right direction.

In this case, determining the polar Bloch points and discussing their distribution just from the polar map may not be accurate. In addition, I saw some typical anti-vortex configurations at h-t-h DWs. Is it reasonable in the planar view of Bloch points system?

Response

Thank you for the careful observation by the reviewer. We carefully checked the marks in Supplementary Fig. 8. We think that the second mark on the fourth line and the mark on the fifth line are divergent polarization patterns, because that these patterns have the polarization vectors with nearly all directions though these patterns are not perfect. The circles might be divergently marked, which affect identification of the reviewer. We modified the Supplementary Fig. 8 with deleted the first circle on the fourth line and moved other two circles on the fourth and fifth lines. This figure is also displayed as follows (Fig. R4).

Fig. R4. The $-\delta\mathbf{r}_i$ vector map showing divergent polarization patterns (black dashed circles) at “tail-to-tail” domain walls (black dashed lines). The arrows with different colors denote the polarization directions of PTO unit cells.

In our work, we first performed the phase-field simulations, which gave the polarization structures of Bloch points. Then, we observed the Bloch points in tensile-strained ultrathin ferroelectric PbTiO_3 films by aberration-corrected scanning transmission electron microscope. The experimental observations are consistent with the simulation results. We think our results is reasonable by both evidence from phase-field simulations and experiment observations. On the other hand, we acquired the low-magnified HAADF-STEM images under defocus mode, which indicates many dots contrast are formed at stripe domain walls. Meanwhile, combining with atomic-scale planar-view observations and the polar map, we fully believe our experiment evidence for polar Bloch points.

The antivortex configurations are common in the planar view. As shown in Fig. R5 [Fig. 2(a-c) in the main text], there exist an antimeron (or antivortex) between two adjacent meron-related Bloch points along both h-t-h and t-t-t domain walls.

Fig. R5. Bloch points associated with antimerons predicted by the phase-field simulations. (a) The horizontal slice of the central layer of the PTO film overlaid with the isosurfaces of $P_{mag} = 0.55 P_s$ (red). The close isosurfaces are associated with BP-M's and the open ones antimerons (or antivortices). (b, c) The zoom-in slices of the two antimerons (or antivortices) marked by dashed boxes in (a). The white and gray arrows show the polarization directions near and away from the antimerons (or antivortices), respectively.

Response to Reviewer #2

Comment #1

The authors need to give a more convincing answer as to WHY such Bloch points form in a uniform PTO layer. It has a metallic SRO bottom electrode as such, at least on one contact electrode they should be efficient screening of the surface charge. So, what is the need to form such exotic Bloch points? Is it possible that their SRO layer does not have the bulk metallicity (or carrier concentration) that results in such exotic domain formation? Some additional information on the metallicity of SRO is needed.

Response

We appreciate this constructive comment regarding the formation mechanism of Bloch points. We believe that the effective screening of the surface charge (or the short-circuit boundary condition) provided by the SRO electrodes is the necessary condition for the formation of Bloch points. To demonstrate that, we have switched the boundary condition to the open-circuit one where a strong depolarization field exists. As shown in Fig. R6, all c domains disappear under the open-circuit boundary condition. The polarization characteristics of Bloch points is that the polarization vectors point to all directions around the core. The out-of-plane polarizations provided by the c domains are the necessary condition for the formation of Bloch points. Under the open-circuit boundary condition, the depolarization field suppresses the c domains and destroys the Bloch point lattice.

As for the metallicity of SRO, previous literatures show the metallic SRO become an insulator

when the film thickness decreases to 2~4 unit cells¹⁻³. The thicknesses of the top and bottom SRO layers in our experiments are both about 5 nm, well above the transition thickness. Thus, our SRO layers are metallic.

Fig. R6. The simulation results under the short-circuit (a) and open-circuit (b) boundary conditions. The c domains shown as red and blue isosurfaces disappear in the open-circuit boundary condition.

Reviewer #3

Comment #1

For magnetic Bloch point, owing to its topology, it is usually associated with the creation or annihilation of a skyrmion (for example, see Fig. 18 in Review of Modern Physics 89, 025006 (2017)). Nevertheless, I didn't see any formation/deformation of polar skyrmion in this study. The authors should give remarks about this point.

Response

Thank you for the comment. We have carefully read this review paper. Fig. 18 described the situation that the merging of two skyrmions is associated with a Bloch point. The creation or annihilation of a skyrmion is also related with Bloch points. However, there are other mechanisms. As an example, Bloch points would emerge in an asymmetric disk during the repeatedly applying and withdrawing of the magnetic field⁴, as shown in Fig. R7.

Fig. R7. Direct observation of stabilized Bloch point (BP) structures⁴. **a–c** Schematic diagram of three possible configurations of a BP with skyrmion charge $q = +1$, namely, hedgehog (a), circulating (b), and spiralling (c) configurations where the colour indicates the orientation of magnetization. **d** In-plane (IP) and out-of-plane (OOP) magnetic components observed by magnetic transmission soft X-ray microscopy, and the corresponding magnetic structures determined by micromagnetic simulations. Zoomed images for magnetic configurations near vortex core structures are also inserted. The black and white contrasts in the images of the IP (OOP) magnetic component represent the magnetizations oriented in the left and right directions on the disk plane (perpendicular upward and downward to the disk plane), respectively. **e** Images of the distorted vortex core structures with no BP (left), a single BP (middle), and double BPs (right), and the configurations of the BPs embedded in the vortex cores. The OOP magnetic components with $m_z > +0.8$ (red) and $m_z < -0.8$ (blue) were extracted and the BP configurations were obtained by interpolation from the simulated vortex core structures. Scale bars in (d, e) correspond to 500 nm and 50 nm, respectively.

Comment #2

In simulation, the authors discovered the Bloch points by leaving an electrode on top of the thin film.

The authors should give an illustration about how this set up looks like in experiment in Fig 1. Also, the authors explored positions of Bloch points by varying the electrode thickness; however, I didn't see what variables electrode thickness correspond to in free energy. This must be mentioned in the method section.

Response

We fully comprehend the concern. In the simulation, we show the Bloch points emerges when an electrode is placed on top of the thin film. The average height of the Bloch points increases with the increase of the thickness of the top electrode (h_{te}) and reaches the middle of the film when h_{te} is larger than 5 nm. Thus, we grew a 5 nm SrRuO₃ electrode on the top of the PbTiO₃ film experimentally.

The total elastic energy density can be written as a piecewise function:

$$f_{elas} = \begin{cases} \frac{1}{2} C_{ijkl} \varepsilon_{ij}(x, y, z) \varepsilon_{kl}(x, y, z), & -h_{be} \leq z < 0 \\ \frac{1}{2} C_{ijkl} [\varepsilon_{ij}(x, y, z) - \varepsilon_{ij}^0(x, y, z)] [\varepsilon_{kl}(x, y, z) - \varepsilon_{kl}^0(x, y, z)], & 0 \leq z \leq h_f \\ \frac{1}{2} C_{ijkl} \varepsilon_{ij}(x, y, z) \varepsilon_{kl}(x, y, z), & h_f < z \leq h_f + h_{te} \end{cases} \quad (1)$$

where h_f , h_{te} and h_{be} are the thicknesses of the ferroelectric film and the top and bottom electrodes, respectively. The total elastic energy F_{elas} is the integral of f_{elas} . In our model, h_f and h_{be} are constant values and only h_{te} varies. As a result, F_{elas} is a function of h_{te} . After solving the mechanical

equilibrium equation $\sigma_{ij} = -\frac{\partial f_{elas}}{\partial P_i}$, the obtained stress σ_{ij} and the mechanical driving force $-\frac{\partial f_{elas}}{\partial P_i} =$

$Q_{ijkl} \sigma_{kl} P_j$ are also functions of h_{te} , which influences the domain structure evolution via the time-dependent Ginzburg-Landau equation.

Change

We have added one sentence (**The introduction of the top electrode influences the mechanical driving force, which will be explained in Supplementary Note 1.**) in the method section on Page 15 in the revised manuscript and added Supplementary Note 1 in the revised supplementary materials.

Comment #3

From Line 79 to Line 84: the authors quickly drew a conclusion that a mechanical force/stress drives the topological transition from meron to Bloch points. It's totally elusive why the mechanical force should be related to electrode thickness and be the decisive factor of topological transition. To me the thickness of electrode is not related to strain at all. Also, I don't understand what a zero thickness of electrode physically corresponds to.

Response

Thank you for the comment. As explained in the Response to Comment #2, the introduction of the top electrode influences the mechanical driving force, which leads to the variation of the domain structure. Thus, it is the mechanical force/stress that drives the topological transition from merons to Bloch points. A zero thickness of the top electrode means there is no top electrode. We added more sentences to make the logic more clearly.

Change

We have added one phrase (**corresponding to the case that there is no top electrode**) and one sentence (**As explained in Supplementary Note 1, the introduction of the top electrode influences the mechanical driving force, which leads to the variation of the domain structure.**) on Page 3 in the revised manuscript.

Comment #4

Figure 2 must be improved. The current version is very confusing. In Fig. 2(a), what does the color denote—polarization magnitude or direction? No colorbar is presented there. I don't understand what those stripes composed of 5 colors stand for. What do red circles represent? I know they are isosurfaces but are they meron-Bloch points? In Fig. 2(l), I cannot differentiate Meron- and Antimeron- Bloch points, though authors claim they correspond to large and small circles but it's very hard to tell from the figure. Are Meron-Bloch points red circles, and Antimeron-Bloch points yellow and purple circles? Are red circles in Fig. 2(l) the same as red circles in Fig. 2(a)?

Response

Thank you for the valuable comment. The color denotes the polarization direction. We have added some labels under Fig. 2(a) to show the relation between the color and the polarization direction. The red circles are isosurfaces of the polarization magnitude $P_{mag} = 0.55 P_s$. The close isosurfaces are associated with meron-Bloch points and most open isosurfaces antimeron-Bloch points. For Fig. 2(l), we now use circles and diamonds to express the meron- and antimeron-Bloch points. The red circles in Fig. 2(l) are the same as red circles in Fig. 2(a). The modified Fig. 2 is presented here as Fig. R8.

Fig. R8. Bloch points associated with antimerons predicted by the phase-field simulations. (a) The horizontal slice of the central layer of the PTO film overlaid with the isosurfaces of $P_{mag} = 0.55 P_s$ (red). **The close isosurfaces are associated with BP-M's and most open isosurfaces BP-AM's.** (b, c) The zoom-in slices of the two antimerons marked by dashed boxes in (a). The white and gray arrows show the polarization directions near and away from the antimerons, respectively. (d-g) The isosurface of $P_{mag} = 0.3 P_s$ (d) and the slices perpendicular to the x (e), y (f) and z (g) axes for the antimeron in (b). (h-k) The isosurface and the slices for the antimeron in (c). The normalized polarization vectors are overlaid therein. (l) The isosurface of $P_{mag} = 0.55 P_s$ (red) where the CCD- and DDC-type Bloch points are labeled by purple and yellow circles, respectively. The circles **and diamonds** represent BP-M's and BP-AM's, respectively. The black arrow labels the abnormal BP-AM and the dashed circles mark the antimerons not associated with Bloch points.

Comment #5

The authors explained why no zero-polarization is observed in experiments by proposing a hypothesis that there is a quadratic-like electric field divergent along z direction. Though by simulation, it can

give resembling polarization patterns to experiments (Fig. S10), I cannot relate the formation of such electric field pattern (Fig. 9b) with any physical charge distribution resulting from work function difference. In my opinion, such explanation does not correspond to any realistic scenario and is not physical.

Response

Thank you for the valuable comment. First, we have to emphasize that we actually observed experimentally the Bloch points, or the zero-polarization states, as shown in Fig. R9(b-e). The difference between the simulations shown in Fig. 2 and the experiments shown in Fig. 3 is that the polarization vectors above and below a BP-AM from the simulations are too short (shown in Fig. R9(f-i)), while those in the experiments are comparable to the bulk polarization (shown in Fig. R9(b-e)). We explained this point more clearly in the revised manuscript.

To make the simulation results more close to the experimental ones, we considered the work function difference between PbTiO_3 (PTO) film grown on the SrRuO_3 (SRO). For a PTO film grown on the SRO electrode, the work function different between the two materials generates a built-in electric field at the PTO/SRO interface, which points from the PTO film to the SRO electrode. As a result, the polarization in the PTO film points to the SRO electrode, as shown in Fig. R10. For the PTO film sandwiched by two SRO electrodes, there exist opposite built-in electric field at the two PTO/SRO interfaces, forming a divergent electric field distribution in the PTO film, as schematically shown in Fig. R11. The explicit expression of the divergent built-in electric field is hard to obtain. In our study, we assume an exponential function as:

$$E_z = E_0 [e^{-z/z_0} - e^{(z-h_f)/z_0}] \quad (2)$$

where E_0 is the maximal electric field at the interface, h_f is the film thickness and z_0 is the characteristic length relating with the decay of the electric field.

Fig. R9. (a) Superposition of reversed Ti-displacement vectors ($-\delta_{Ti}$) and atomic-resolved HAADF-STEM image. The $-\delta_{Ti}$ vectors of PTO unit cells were shown as yellow arrows. (b-e) Magnified $-\delta_{Ti}$ vector maps showing four typical polarization patterns corresponding to the four red boxes in (a). The location of Bloch points are marked by dashed circles in (b-e). (f-i) The simulation results without the out-of-plane divergent electric field. The polarization is very small along the column through the BP's.

Fig. R10. The polarization direction of the PTO film grown on the SRO electrode⁵. In the as-grown state (b), the polarization of the PTO film points from PTO to SRO.

Fig. R11. The schematic diagram of the built-in electric field at the two PTO/SRO interfaces.

Change

To avoid the misunderstanding. We modified the clause “**which contradicts with the experimental observation**” as one sentence “**However, the polarization vectors around the BP-AM’s in experiments are comparable to the bulk polarization**” on Page 8 in the revised manuscript.

To better understand the divergent built-in electric field, we inserted Fig. R11 in Supplementary Fig. 9(a) in the revised supplementary materials. We also added one sentence “**The profile of the electric field is assumed to be $E_z = E_0[e^{-z/z_0} - e^{(z-h_f)/z_0}]$, where E_0 is the maximal electric field at the interface, h_f is the film thickness and z_0 is the characteristic length relating with the decay of the electric field.**” in the caption of Supplementary Fig. 9.

Comment #6

Considering there is no zero-polarization region observed in experiments, I am wondering if there is any other experimental evidence to support the emergence of Bloch points, in addition to Fig. 3(b). For example, what would be the other orthogonal cross-sectional view of Section (I)-(IV)? Does that agree with calculation in Figs. 1 and 2?

Response

Thank you for the valuable comment. As shown in the Response to Comment #5, we have already observed the zero-polarization region in experiments.

We also perform the observation from the other orthogonal cross-sectional view, as seen in Fig. R12. It is seen that the areas “1” and “3” are two typical antivortex patterns, while the areas “2” exhibit a divergent polarization pattern, which is similar to the observation from orthogonal in-plane [001] direction of SSO (Supplementary Fig. 11).

Fig. R12. (a) The low-magnification high-resolution HAADF-STEM image taken along orthogonal $[1\bar{1}0]$ direction of SSO. (b) Superposition of reversed Ti-displacement vectors ($-\delta_{Ti}$) and atomic-resolved HAADF-STEM image. The $-\delta_{Ti}$ vectors of PTO unit cells were shown as yellow arrows. (c-e) Magnified $-\delta_{Ti}$ vector maps showing three typical polarization patterns corresponding to the three white boxes in (a).

Comment #7

On Line 177: “all antimerons transform into DDC-type Bloch points”. Is it CCD instead? Fig. 3c (and Line 150) shows they are divergent, while Line 180 mentions it as “convergent”. This is really confusing. I also don’t understand conversions of DDC to CCD Bloch points shown in Fig. S9(e)(f); I didn’t see a drastic change of the numbers of yellow (or purple) circles, while the authors claimed one type has transformed to the other.

Response

Thank you for the valuable comment. Indeed, all antimerons transform into DDC-type Bloch points. Due to the divergent electric field, the polarization in the z direction is divergent. The polarization in the xy plane is half-divergent-half-convergent. Thus, the polarization is divergent in two directions and convergent in one direction, i.e., the DDC-type.

The “convergent” in Line 180 is a typo. It should be “divergent”.

In the old version of Fig. S9, all Bloch points are marked, so that the change of colors are not

obvious. In the revised version (Fig. R13), only the Bloch points associated with antivortices are marked. It is found that all purple diamonds change to the yellow ones.

We have also shown the variation of the polarization structure after the application of divergent electric field.

Fig. R13. The isosurface of $P_{mag} = 0.55 P_s$ (red) before (e) and after (f) the application of the electric field. CCD- and DDC-type Bloch points are marked by purple and yellow diamonds, respectively. Only BP-AM's are marked for a better visual effect. The detailed variation of two representative antimerons are marked by two dashed boxes and shown in Fig. R14.

Fig. R14. The variation of two antivortices after the effect of electric field. (a, b) and (c, d) correspond to the dashed boxes “1” and “2”, respectively. (a) and (c) are the initial states and (b) and (d) are the state under the electric field. The left, middle and right columns are the slices perpendicular to the x , y and z axes, respectively.

Change

We have replaced Supplementary Fig. 9(e, f) with Fig. R13(a, b) and added Fig. R14 as Supplementary Fig. 10 in the revised supplementary materials.

Comment #8

On Line 103, 104: the authors mention “close” and “open” isosurfaces. I assume the authors refer to the geometry shown in Figs. 1(d)(e) and Figs. 2(d)(h); the authors should give clear indications about these terminologies.

Response

Thank you for the valuable comment. We have marked “close” and “open” on Fig. 2(a) to make them more clearly. The revised Fig. 2(a) is shown here as Fig. R15.

Fig. R15. The horizontal slice of the central layer of the PTO film overlaid with the isosurfaces of $P_{mag} = 0.55 P_s$ (red) where two typical antimerons are marked by dashed boxes. The close and open isosurfaces as marked are associated with BP-M and BP-AM, respectively.

Comment #9

On Line 130: I assume the PTO is grown along [001] direction. The authors should specify the growth direction.

Response

Thank you for the valuable comment. The PTO is grown along [001] direction.

Change

We have added a phrase “(001)_{pc}-oriented” on Page 2 in the revised manuscript.

Comment #10

In Fig. 3e, the convergent polarization patterns don't form in a periodic pattern; I don't understand

why the authors would like to call it an “approximated lattice.”

Response

Thank you for the valuable comment. The convergent polarization patterns are arranged along each h-t-h DW, forming a one-dimensional convergent polarization pattern array, as shown in Fig. 3e. Similarly, the divergent polarization patterns are arranged along each t-t-t DW, forming a one-dimensional divergent polarization pattern array. Indeed, the interspacing between convergent (divergent) polarization patterns is not uniform. However, the h-t-h DWs and t-t-t DWs are arranged periodically. We acquired the planar-view HAADF-STEM image, as shown in Fig. R16. The periodicity is consistent of the width of stripe a_1/a_2 domains, which is about 10 nm, as shown in the inset of Fig. R16. Thus, the convergent polarization pattern arrays and divergent polarization pattern arrays are alternately arranged with periodicity of 10 nm.

Fig. R16. A low-magnification planar-view dark-field TEM image exhibits the regular stripe a_1/a_2 -like domains. The inset is the magnified image corresponding to the area marked by a white box, which indicates that the width of these stripe a_1/a_2 -like domains is about 10 nm.

Change

To make the statement clearly, we have modified the main text as seen in **RED** on Page 8, as “Importantly, at **each** h-t-h DW (labeled by black solid lines in Fig. 3e), many convergent polarization patterns (labeled by black solid circles in Fig. 3e) exist, approximately forming **one-dimensional array**. It is noted that the stripe domains are arranged periodically with periodicity of 10 nm, as shown in Supplementary Fig. 4. Thus, the periodical arrangement of convergent polarization pattern arrays at h-t-h DWs are formed, as shown in Fig. 3e. The periodicity is consistent of the width of stripe a_1/a_2 domains. A typical area in Fig. 3e is selected and its zoom-in polarization map is shown in Fig. 3f, showing a convergent polarization configuration. On the other hand, many divergent polarization

patterns are observed at t-t-t DWs, which also form one-dimensional arrays with **periodical arrangement** (Supplementary Fig. 8).”

Comment #11

The logic from local zero polarization to negative capacitance is not explicit; the authors should make the transition smoother there.

Response

Thank you for the valuable comment. As shown in Fig. R17, the free energy of a typical ferroelectrics is in the shape of a double well. In the state of zero polarization, the curvature of the free energy with respect to the polarization, which is the reciprocal dielectric constant ϵ^{-1} , is negative.

Fig. R17. The bulk free energy density as the function of polarization for ferroelectric PbTiO₃. The red and blue regions show the positive and negative capacitances.

Change

We have added several sentences (As shown in Supplementary Fig. 13, the free energy of a typical ferroelectrics is in the shape of a double well. In the state of zero polarization, the curvature of the free energy with respect to the polarization, which is the reciprocal dielectric constant ϵ^{-1} , is negative.) on Page 9-10 in the revised manuscript and added Fig. R17 as Supplementary Fig. 13 in the revised Supplementary materials.

Comment #12

How to calculate local D in phase-field simulation? Is it to calculate local P and E and then get D from $D = \epsilon_0 E + P$? This point should be made clear in the method section.

Response

Thank you for the valuable comment. The formula to calculate local D is given in this version.

Change

We have added one sentence (**The local electric displacement (D_z) is calculated as $D_z = \epsilon_0\epsilon_b E_z + P_z$.**) on Page 15 in the revised manuscript.

Comment #13

Figures similar to Figs. 4(b)-(f) have been presented in the main manuscript; I would suggest swapping them with Figs. S12(b)(c)(e)(f)(g)(h) in the main manuscript.

Response

Thank you for the comment. Although Figs. 4(b)-(f) are similar to Figs. S11(a)(c)-(f), the two sets of figures are about the electric field and polarization distributions around the DDC- and CCD-type Bloch points. So, all of them should be kept in the manuscript and the supplementary materials. Figs. S12(b)(c)(e)(f)(g)(h) are about the bound charge analysis, which is used to explain the antiparallel distribution of the electric field. In previous literatures^{6,7}, the most direct way to show the negative capacitance is the comparison of the electric field and polarization distributions, as shown in Fig. R18. Thus, we prefer not to modify Fig. 4.

Fig. 3 | Measurement of local electric field and polarization field using EMPAD-STEM. **a**, Polarization vector map from a sub-region of a PbTiO₃ layer embedded within a (SrTiO₃)₁₂/(PbTiO₃)₁₂ superlattice as measured using STEM (details in Methods, Extended Data Figs. 1 and 2). **b**, Local electric field in a PbTiO₃ layer (corresponding to the same region shown in **a**) as measured using TEM. **c**, Variation in the *z* components of local polarization (*P_z*; red hexagons) and electric field (*E_z*; blue circles) along a horizontal line (indicated by the horizontal lines in **a** and **b**) that passes through the core of the vortices. **d**, Local energy density estimated from the variation in *P_z* and *E_z* along the same line. Regions around the core (arrowed) have negative curvature ($\partial^2 G / \partial D^2 < 0$). See details in Methods sections 'Estimation of the free energy (*G*)' and 'Estimation of the permittivity'.

Fig. 3 | Measurement of local polarization, electric field and local potential energy of the polar skyrmion using SCBED. **a**, A cross-sectional high-angle annular dark-field STEM image of the skyrmions in the PbTiO₃ layer. **b, c**, Polarization (**b**) and electric field (**c**) vector maps of the cross-section geometry (*x-z* plane) of the skyrmion measured using SCBED. We can access the Néel (blue box) and Bloch (red box) components from the skyrmion cross-section. The out-of-plane polarizations are separated by in-plane Bloch chiral domain walls (dark regions). The colour wheel hue (saturation) in **c** corresponds to the direction (magnitude) of the in-plane component of the ferroelectric polarization. **d**, Variation of the *z* components of the local polarization (*P_z*; blue curve) and electric field (*E_z*; orange curve) along a horizontal line drawn through the centre of a polar skyrmion, indicated by the dashed lines in **b** and **c**. **e**, Local potential energy estimated from the variation in *P_z* and *E_z* along the same line. Regions around the skyrmion walls (indicated by arrows) have a local energy higher than the surroundings with a negative curvature ($\partial^2 G / \partial P^2 < 0$), indicating local negative permittivity.

Fig. R18. The electric field and polarization distributions reported in previous literatures^{6,7} to illustrate the negative capacitance.

References

- 1 Toyota, D. *et al.* Thickness-dependent electronic structure of ultrathin SrRuO₃ films studied by in situ photoemission spectroscopy. *Appl. Phys. Lett.* **87**, 162508 (2005).
- 2 Chang, Y. J. *et al.* Fundamental thickness limit of itinerant ferromagnetic SrRuO₃ thin films. *Phys Rev Lett* **103**, 057201 (2009).
- 3 Xia, J., Simons, W., Koster, G., Beasley, M. R. & Kapitulnik, A. Critical thickness for itinerant ferromagnetism in

ultrathin films of SrRuO₃. *Physical Review B* **79**, 140407 (2009).

- 4 Im, M.-Y. *et al.* Dynamics of the Bloch point in an asymmetric permalloy disk. *Nature Communications* **10**, 593 (2019).
- 5 Li, M. *et al.* Electric-field control of the nucleation and motion of isolated three-fold polar vertices. *Nat Commun* **13**, 6340 (2022).
- 6 Yadav, A. K. *et al.* Spatially resolved steady-state negative capacitance. *Nature* **565**, 468-471 (2019).
- 7 Das, S. *et al.* Local negative permittivity and topological phase transition in polar skyrmions. *Nature Materials* **20**, 194-201 (2021).

REVIEWER COMMENTS

Reviewer #1 (Remarks to the Author):

Thank the authors for providing sufficient experiments and simulations for my questions and concerns. I believe that all the evidences are robust and adequate for supporting the conclusions. Exploring Bloch points and approaching a realistic negative capacitance in ferroelectrics are undebatable important. Therefore, I recommend this work publication in Nature Communications.

Reviewer #3 (Remarks to the Author):

Please see attachment.

The authors have addressed many of my concerns. However, I still have some remaining doubts for the authors to further clarify:

1. In the Comment #2 and the Method section and the Supplementary Note I, the authors mentioned that a mechanical equilibrium condition $\sigma_{ij,j} = 0$ needs to be solved. However, I didn't see the definition for $\sigma_{ij,j}$. I assume the subscript means a derivative of P_j to the second-rank stress tensor σ_{ij} , but I don't understand why $\sigma_{ij,j} = 0$ corresponds to a mechanical equilibrium condition, given $-\frac{\partial f_{elas}}{\partial P_i} = Q_{ijkl}\sigma_{kl}P_j$. Similar explanation about $D_{i,i} = 0$ should also be given. It also confuses me that in Methods part, Q_{ijkl} relates the spontaneous strain with $\varepsilon_0 = Q_{ijkl}P_kP_l$, while in supplementary note and rebuttal letter, $-\frac{\partial f_{elas}}{\partial P_i} = Q_{ijkl}\sigma_{kl}P_j$. The authors should make relations between ε (ε_0), Q , P , f_{elas} consistent and more explicit.
2. I don't believe the divergent electric field model proposed in comment #5. If SrRuO₃ are top and bottom electrodes, how can electric field keep increasing (and increase so sharply) when it goes from interface to the deeper region of the electrode? In the deeper region of the electrode, electric fields should become zero due to the screening effect.
3. In Comment #6, does Fig. S11 correspond to in-plane view of [001]? I thought [001] is the film growth out-of-plane direction?
4. In Comment #11, the double well potential picture is usually for a 1D displacive ferroelectric phase transition: there $P = 0$ corresponds to the saddle point, and two local minima corresponds the situation where all polarizations point along the same direction. Nevertheless, I doubt whether this saddle point can be applied to a Bloch point, characterized by divergent or convergent polarizations from all directions.

Without addressing these questions properly, I would not recommend this article to be published as Nat. Comm.

Response to Reviewers' Comments:

Ref: NCOMMS-23-27559A

Title: Polar Bloch points in strained ferroelectric films

11 March 2024

We appreciate the recommendation of publication raised by reviewer #1. We are also grateful to reviewer #3 for that “The authors have addressed many of my concerns”. Nevertheless, reviewer #3 remains some doubts to be further clarified. We fully understand the concerns by the reviewer, and we have addressed all the concerns point-to-point in the following. Key revisions are highlighted in **RED** in the revised manuscript.

Reviewer #3

Comment #1

In the Comment #2 and the Method section and the Supplementary Note I, the authors mentioned that a mechanical equilibrium condition $\sigma_{ij,j} = 0$ needs to be solved. However, I didn't see the definition for $\sigma_{ij,j}$. I assume the subscript means a derivative of P_j to the second-rank stress tensor σ_{ij} , but I don't understand why $\sigma_{ij,j} = 0$ corresponds to a mechanical equilibrium condition, given $-\frac{\partial f_{elas}}{\partial P_i} = Q_{ijkl}\sigma_{kl}P_j$. Similar explanation about $D_{i,i} = 0$ should also be given. It also confuses me that in Methods part, Q_{ijkl} relates the spontaneous strain with $\varepsilon_0 = Q_{ijkl}P_kP_l$, while in supplementary note and rebuttal letter, $-\frac{\partial f_{elas}}{\partial P_i} = Q_{ijkl}\sigma_{kl}P_j$. The authors should make relations between ε (ε_0), Q , f_{elas} consistent and more explicit.

Response Comment #1:

We appreciate the concerns raised by the reviewer, which remind us that some expressions should be more explicit. $\sigma_{ij,j}$ means the spatial derivation of σ_{ij} , or $\frac{\partial \sigma_{ij}}{\partial x_j}$. By solving the mechanical equilibrium equation, $\sigma_{ij,j} = 0$, the stress tensor σ_{ij} is obtained. Substituting $f_{elas} = \frac{1}{2}C_{klmn}(\varepsilon_{kl} - \varepsilon_{kl}^0)(\varepsilon_{mn} - \varepsilon_{mn}^0)$ to the mechanical driving force $-\frac{\partial f_{elas}}{\partial P_i}$, one can obtain

$$-\frac{\partial f_{elas}}{\partial P_i} = \frac{1}{2}C_{klmn} \left[\frac{\partial \varepsilon_{kl}^0}{\partial P_i} (\varepsilon_{mn} - \varepsilon_{mn}^0) + (\varepsilon_{kl} - \varepsilon_{kl}^0) \frac{\partial \varepsilon_{mn}^0}{\partial P_i} \right] = C_{klmn} \frac{\partial \varepsilon_{kl}^0}{\partial P_i} (\varepsilon_{mn} - \varepsilon_{mn}^0) = \sigma_{kl} \frac{\partial \varepsilon_{kl}^0}{\partial P_i} \quad (1)$$

Since $\varepsilon_{kl}^0 = Q_{klst}P_sP_t$, $\frac{\partial \varepsilon_{kl}^0}{\partial P_i} = Q_{klit}P_t + Q_{klis}P_s = 2Q_{klit}P_t = 2Q_{ijkl}P_j$. Finally,

$$-\frac{\partial f_{elas}}{\partial P_i} = 2Q_{ijkl}\sigma_{kl}P_j \quad (2)$$

It is worthwhile to mention that in the previous version there was a typo by missing the factor 2, although this did not influence the program.

Changes

To make the statement more clearly, we have complemented the equations containing the commas on Page 14 of the main text (where $P_{i,j} = \frac{\partial P_i}{\partial x_j} \dots \sigma_{ij,j} = \frac{\partial \sigma_{ij}}{\partial x_j} = 0 \dots D_{i,i} = \frac{\partial D_i}{\partial x_i} = 0$). We have also modified several sentences and added the deduction of the mechanical driving force on Page 2 of the Supplementary materials.

Comment #2

I don't believe the divergent electric field model proposed in comment #5. If SrRuO₃ are top and bottom electrodes, how can electric field keep increasing (and increase so sharply) when it goes from interface to the deeper region of the electrode? In the deeper region of the electrode, electric fields should become zero due to the screening effect.

Response to Comment #2:

We appreciate the concerns raised by the reviewer, which remind us that the plotting in the previous version should be more clearly presented. Actually, the electric field only exists in the PbTiO₃ film and the electric field in the SrRuO₃ electrode is zero. In this new manuscript, we have removed the pair of black arrows in Fig. S9(a) in order to avoid any possible misunderstanding. The revised Fig. S9(a) is also presented here as Fig. R1. The electric field vectors in the PbTiO₃ film always point towards the PbTiO₃/SrRuO₃ interfaces due to the work function difference between the two materials. Since the work function difference is an interfacial effect, the electric field magnitude should be the largest at the interfaces. That is the reason why we assume an exponential function for the electric field.

Fig. R1. The schematic distribution of the electric field due to the work function difference between SrRuO₃ and PbTiO₃.

Comment #3

In Comment #6, does Fig. S11 correspond to in-plane view of [001]? I thought [001] is the film growth out-of-plane direction?

Response to Comment #3:

We fully understand the concerns by the reviewer. The [001] direction is in the orthorhombic crystal system. The out-of-plane direction is the [110] direction. The relationship between the Miller indices in the orthorhombic and pseudocubic crystal systems is: $[110]_O // [001]_{PC}$, $[1\bar{1}0]_O // [100]_{PC}$ and $[001]_O // [010]_{PC}$. The subscripts “O” and “PC” represent the orthorhombic and pseudocubic crystal system, respectively.

Changes

We have modified the expression “(001)_{pc}-oriented” as “orthorhombic (110)-oriented [or the pseudocubic (001)-oriented]” on Page 2 of the main text to make it more clear.

Comment #4

In Comment #11, the double well potential picture is usually for a 1D displacive ferroelectric phase transition: there $P = 0$ corresponds to the saddle point, and two local minima corresponds the situation where all polarizations point along the same direction. Nevertheless, I doubt whether this saddle point can be applied to a Bloch point, characterized by divergent or convergent polarizations from all directions.

Response to Comment #4:

We appreciate the comments by the reviewer. The purpose of the double well potential picture is to show the relation between the $P=0$ state and the negative capacitance. It is indeed a model case, suitable for the 1D ferroelectric system. For more complex cases, such as the negative capacitance in polar topological structures, the common method is to extract the polarization and electric field curves and demonstrate that they show opposite trends. For example, the polarizations around a vortex and a skyrmion show the swirling pattern, or the polarizations orient in all directions in planes cutting the vortex and skyrmion. The local negative capacitance in the vortex core and skyrmion periphery is determined by the opposite trends of the line profiles of the polarization and electric field, as shown in Fig. R2. In the case of Bloch points, we also found the polarization and electric field show opposite trends across the Bloch point. Moreover, we directly calculated the dielectric constant and found this quantity show negative values around the Bloch point. In other words, the local negative capacitance around the Bloch points is demonstrated.

Fig. 3 | Measurement of local electric field and polarization field using EMPAD-STEM. **a**, Polarization vector map from a sub-region of a PbTiO₃ layer embedded within a (SrTiO₃)₁₂/(PbTiO₃)₁₂ superlattice as measured using STEM (details in Methods, Extended Data Figs. 1 and 2). **b**, Local electric field in a PbTiO₃ layer (corresponding to the same region shown in **a**) as measured using TEM. **c**, Variation in the *z* components of local polarization (*P_z*; red hexagons) and electric field (*E_z*; blue circles) along a horizontal line (indicated by the horizontal lines in **a** and **b**) that passes through the core of the vortices. **d**, Local energy density estimated from the variation in *P_z* and *E_z* along the same line. Regions around the core (arrowed) have negative curvature ($\partial^2 G / \partial D^2 < 0$). See details in Methods sections 'Estimation of the free energy (*G*)' and 'Estimation of the permittivity'.

Fig. 3 | Measurement of local polarization, electric field and local potential energy of the polar skyrmion using SCBED. **a**, A cross-sectional high-angle annular dark-field STEM image of the skyrmions in the PbTiO₃ layer. **b, c**, Polarization (**b**) and electric field (**c**) vector maps of the cross-section geometry (*x-z* plane) of the skyrmion measured using SCBED. We can access the Neel (blue box) and Bloch (red box) components from the skyrmion cross-section. The out-of-plane polarizations are separated by in-plane Bloch chiral domain walls (dark regions). The colour wheel hue (saturation) in **c** corresponds to the direction (magnitude) of the in-plane component of the ferroelectric polarization. **d**, Variation of the *z* components of the local polarization (*P_z*; blue curve) and electric field (*E_z*; orange curve) along a horizontal line drawn through the centre of a polar skyrmion, indicated by the dashed lines in **b** and **c**. **e**, Local potential energy estimated from the variation in *P_z* and *E_z* along the same line. Regions around the skyrmion walls (indicated by arrows) have a local energy higher than the surroundings with a negative curvature ($\partial^2 G / \partial P^2 < 0$), indicating local negative permittivity.

Fig. R2. The electric field and polarization distributions reported in previous literatures^{1,2} to illustrate the negative capacitance.

- 1 Yadav, A. K. *et al.* Spatially resolved steady-state negative capacitance. *Nature* **565**, 468-471 (2019).
- 2 Das, S. *et al.* Local negative permittivity and topological phase transition in polar skyrmions. *Nature Materials* **20**, 194-201 (2021).

REVIEWERS' COMMENTS

Reviewer #3 (Remarks to the Author):

Please see attachment.

I am satisfied with the authors' revision, and I recommend the current version of the manuscript to be published as a Nat. Comm. Two minor comments are suggested as below and I don't need to see the revised manuscript again:

1. Maybe the authors should specify the thickness of PTO in Fig. S9. According to my understanding, in Fig. S9(b), h_f is 5 nm.
2. I've noticed in a recent publication Phys. Rev. Lett. **132**, 026902 (2024), ferroelectric Bloch point is also observed, though there the nature of the Bloch point is dynamical. The authors should also cite this work in the introduction, considering the discovery of ferroelectric Bloch point is rare in recent studies.

Response to Reviewers' Comments:

Ref: NCOMMS-23-27559B

Title: Polar Bloch points in strained ferroelectric films

9 April 2024

We appreciate the recommendation of publication raised by reviewer #3. We fully understand the concerns by the reviewer, and we have addressed all the concerns point-to-point in the following. Key revisions are highlighted in **RED** in the revised manuscript.

Reviewer #3

Comment #1

Maybe the authors should specify the thickness of PTO in Fig. S9. According to my understanding, in Fig. S9(b), h_f is 5 nm.

Response Comment #1:

We appreciate the careful inspection by the reviewer. h_f is 5 nm. We have added this information in the legend of Supplementary Fig. 9. We also specified the values of E_0 (9.1 MV/cm) and z_0 (0.8 nm).

Comment #2

I've noticed in a recent publication Phys. Rev. Lett. **132**, 026902 (2024), ferroelectric Bloch point is also observed, though there the nature of the Bloch point is dynamical. The authors should also cite this work in the introduction, considering the discovery of ferroelectric Bloch point is rare in recent studies.

Response to Comment #2:

Thank you for the reminder. We have cited this paper in the revised manuscript.

Changes

We have added one sentence “**Recently, Gao et al. used the effective Hamiltonian approach to study the evolution of ultrathin Pb(Zr_{0.4}Ti_{0.6})O₃ film under an optical vortex beam and found the skyrmion state could be generated, accompanied with the emergence of a Bloch point³⁶.**” on Page 2 in the main text to cite this paper.